# CALLME: Call Graph Augmentation with Large Language Models for Javascript

**Michael Wang**
MIT CSAIL
Cambridge, MA 02139, USA
mi27950@mit.edu

**Kexin Pei**
Department of Computer Science
University of Chicago
Chicago, IL 60637, USA
kpei@uchicago.edu

**Armando Solar-Lezama**
MIT CSAIL
Cambridge, MA 02139, USA
asolar@csail.mit.edu

## Abstract

Building precise call graphs for Javascript programs is a fundamental building block for many important software engineering and security applications such as bug detection, program repair, and refactoring. However, resolving dynamic calls using static analysis is challenging because it requires enumerating all possible values of both the object and the field. As a result, static call graph construction algorithms for Javascript ignore such dynamic calls, resulting in missed edges and a high false negative rate. We present a new approach, CALLME, that combines Language Models (LMs) with a custom static analyzer to address this challenge. Our key insight is in using LMs to incorporate additional modalities such as variable names, natural language documentation, and calling contexts, which are often sufficient to resolve dynamic property calls, but are difficult to incorporate in traditional static analysis. We implement our approach in CALLME and evaluate it on a dataset of call edges that are dependent on dynamic property accesses. CALLME achieves 80% accuracy and .79 F1, outperforming the state-of-the-art static analyzer by 30% and .60, respectively. To study the effectiveness of CALLME on downstream analysis tasks, we evaluate it on our manually curated dataset with 25 known Javascript vulnerabilities. CALLME can detect 24 vulnerabilities with only 3 false positives, whereas static analysis tools based on current call graph construction algorithms miss all of them.

## 1 Introduction

Call graph construction has been a prerequisite for many critical software analysis tasks, such as code optimization (Fink et al., 2008; Malavolta et al., 2023), bug detection (Brown et al., 2017; Cai et al., 2023a; 2021), taint analysis (Kang et al., 2023), and software maintenance and inspection (Feldthaus et al., 2013). However, dynamic languages such as Javascript present unique challenges for call graph construction. The dynamic nature of Javascript alongside the size of sophisticated frameworks such as React and AngularJS makes call graph construction very difficult. For example, a recent survey on Javascript call graph construction (Antal et al., 2023) has shown that even the best static analyzer only obtains a 0.43 detection F1 score, suffering from high false positives and false negatives. Moreover, the majority of the existing static analyzers tested simply ignore the method calls when they are across multiple files (Antal et al., 2023).

A key challenge for constructing call graphs statically is Javascript's flexible object model, which allows properties to be created and deleted at runtime. Specifically, dynamic property accesses, where the property being accessed depends on a runtime-computed string, are esti-

mated to cause 70% of missed edges in static call graph construction algorithms (Chakraborty et al., 2022). Recent static call graph construction algorithms specifically ignore reasoning about most dynamic property accesses due to their runtime-dependent behavior, and thus miss any calls that are computed dynamically from objects (Feldthaus et al., 2013; Nielsen et al., 2021a).

Figure 1 shows an example of a dynamic property access, where the property access call is made in line 13 to a function field on the object `o`. Analyzing the call requires a field-sensitive pointer analysis to determine the points-to set of `o["greet"+firstUser]`. However, the property names of `o` are computed dynamically in a loop in lines 3-8. Computing a field-sensitive pointer analysis with the presence of dynamic property accesses is thus a prohibitively expensive process. The presence of dynamic properties increase pointer analysis runtime from $O(N^3)$ to $O(N^4)$, where $N$ is the size of the program (Sridharan et al., 2012). As a result, traditional field-sensitive analyses are unable to handle large Javascript frameworks such as *jQuery* and *react*.

```
1   var o = {};
2   function createGreetFunctions() {
3     for (user of this.getUsers()) {
4       var greetUser = "greet" + user;
5       o[greetUser] = new function(){
6         console.log(arguments[0] + \\
7                     " " + user);
8       }
9     }
10  }
11  createGreetFunctions();
12  var firstUser = this.getUsers()[0];
13  o["greet" + firstUser]("hello");
```

Figure 1: A basic example of a dynamic property access function call on line 13. The value of `firstUser` is determined at runtime, making it difficult for static analyzers.

In this paper, we introduce CALLME, an approach to specifically target dynamic property access calls for Javascript call graph construction. CALLME has two stages: statement selection and inference.

1. **Statement selection.** We develop JSelect, a custom static analyzer that efficiently selects the relevant statements needed to determine if a call site calls a specified function.

2. **Inference.** The output of JSelect is further processed by a Language Model (LM) which determines whether the call site can call the function.

Importantly, CALLME does not replace the need for traditional call graph construction algorithms. Rather, CALLME is intended to augment traditional call graph construction specifically for dynamic property accesses. To the best of our knowledge, CALLME is the first static analysis system capable of resolving dynamic property accesses without runtime information.

Reasoning about dynamic property accesses for Javascript makes a good target for LMs for several reasons. First, existing solutions are already unsound and incomplete, so a solution that works well in practice can be competitive even without theoretical guarantees – introducing an unsound LM does not worsen any existing guarantees. Second, LMs can take advantage of useful sources of information that are difficult to encode as traditional static analysis rules. In Figure 1, for example, it is fairly straightforward to infer that the function call made in line 13 likely refers to the function defined in line 5, given the variable names such as "greet" and "User". Similarly, inferring the relationship between words such as "greet" and "hello" often relies on knowledge of natural language. Additionally, LMs can infer high-level design patterns, such as visitors and builders, which can be challenging to incorporate into traditional static analysis.

We evaluate CALLME on a dataset of caller-callee edges collected by dynamic analysis (Chakraborty et al., 2022), where we specifically target dynamic property accesses. Importantly, the dataset consists only of calls that were missed by Approximate Call Graph (ACG) (Feldthaus et al., 2013), a recent call graph construction algorithm. CALLME is able to resolve 75% of calls with an F1 score of 0.79. Jelly, the recent open source implementation based off of the state-of-the-art static analysis framework JAM (Nielsen et al., 2021a), is only able to resolve 11% of these calls with an F1 score of .19 by handling dynamic property accesses that start or end with hard-coded strings, *e.g.,* `obj["foo" + y]()`.

We show how CALLME can be applied downstream program analysis tasks such as vulnerability detection by resolving edges that are currently undetectable by call graph construction algorithms due to their use of dynamic property accesses. We manually searched multiple datasets of known Javascript bugs in real-world projects and identified 25 function calls across 23 projects resulting in bugs or vulnerabilities that are undetectable with current call graph construction due to dynamic property accesses. These security bugs include prototype pollution, cross-site scripting, command injection, and arbitrary file overwrites. We build a scanner using CALLME to search for these vulnerable function calls, which resolves 24 out of 25 vulnerable calls with only 3 false positives.

## 2 Motivation

As an example to motivate our approach, consider a simplified code snippet (Figure 2) from the jQuery framework (openjsf.org) containing a call that is ignored by static analyzers. In the following, we discuss several features of Javascript that make analyzing the call difficult.

**Functions are objects.** Functions are objects which can have assigned fields themselves. This is shown on line 9, where the initial jQuery object is a function but also has a field called extend. Functions can also be passed around as objects, as shown in the each function on line 28.

**Dynamic additions/uses.** Properties and fields can be added to objects dynamically, such as adding extend to the jQuery object on line 9, and then using extend to add the each and show properties. The properties can then be overwritten, as shown on line 30.

**Arity mismatching.** Functions can be called with any number of arguments, as shown in the extend function on lines 8 and 10.

**Computed names.** Properties can be read and written by computed names, as seen on line 29. Additionally, precise modeling of functions is necessary for an analyzer. In our example, an analyzer would need to model the functionality of each in order to understand what happens on line 28, and would need to model the functionality of extend to understand what happens on line 17. Without the explicit modeling of extend, there is no way for a static analyzer to know that the showAll and each functions are added to jQuery itself.

However, for a human looking at the code snippet in Figure 2, there are several things that indicate that the function call cssFn.apply(this,arguments) on line 32 can refer to the function showAll on line 23. There are natural language hints and

```
1   jQuery = function(selector, context); {
2       return new jQuery.prototype.init();
3   }
            1.Functions are objects.
4
5
6       2.Dynamic additions/uses.
7                           3.Arity mismatching.
8
9   jQuery.extend  =  function()  {
10      for (i=0; i < arguments.length; i++ ) {
11          for ( name in  arguments[i]  ) {
12              this[name] = options[name];
13          }
14      }
15  };
16
17  jQuery.extend ( {
18      each : function( obj, cb ) {
19          for ( i in obj ) {
20              cb.call(obj[i], i, obj[i]);
21          }
22      },
23      showAll :  function()  {
24          return showHide( this, true );
25      }
                              4.Computed names.
26  });
27
28  jQuery.each (["show"],function(i,name){
29      var cssFn =  jQuery.fn[name + "All"] ;
30      jQuery[name] = function(speed) {
31          return speed == null ?
32              cssFn.apply(this,arguments)  :
33              this.animate(name, speed);
34      };
                  Dynamic property access call.
35  });
```

Figure 2: A modified code snippet taken from jQuery showing several features of Javascript that make static analysis difficult. The dynamic property access is on line 29, and the function call is on line 32. However, information such as the variable names allow an LM to successfully resolve the call.

background knowledge of programming patterns that are helpful. For example, the anonymous function defined on line 28 takes two parameters, i and name, a common pattern where

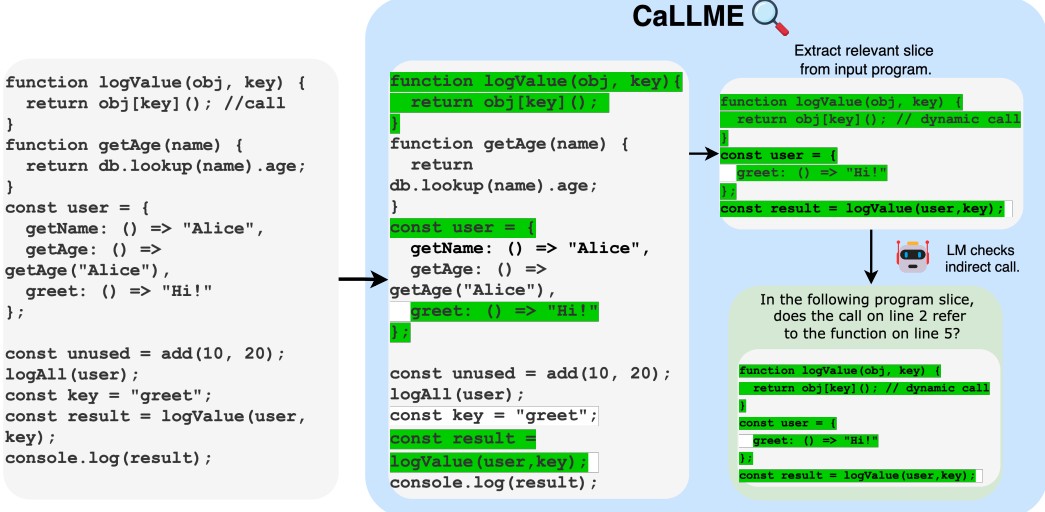

Figure 3: An overview of our system CALLME. JSelect selects the relevant statements from the original program, which then gets formulated into a prompt for an LM.

i refers to an index and name refers to the value of a list. each is a word commonly used in programming when iterating over lists, so it is fairly easy to infer that jQuery.each is iterating through each element in the list of its first argument, even without looking at the definition of each on line 18. While these types of heuristics are very difficult to encode in a traditional static analyzer, they can be incorporated into machine learning models and large language models. CodeLlama-34B-Instruct, a 34-billion parameter large language model optimized for code generation and understanding released by Meta AI in 2023 (Rozière et al., 2024a), successfully resolves the call on line 32.

## 3 Approach

CodeLlama-34b is able to identify the call in Figure 2 because of its small size at 35 lines. In a real-world program, the call site and target function could be several thousand lines apart, with most of the code being irrelevant to the function call we are trying to resolve. Thus, we need a way to automatically find the relevant lines to feed into the LM. We implement a custom statement selection algorithm, JSelect, which outputs the relevant lines to be formulated into a prompt for an LM. An overview of CALLME can be seen in Figure 3.

```
1   can.each(options, function(attr) {
2       options.attr = attr;
3       if ( Scanner.attributes [attr]) {
4           Scanner.attributes [attr](options, el);
5       };
6   });           Global variables are defined elsewhere.
```

Figure 4: We want to provide local information around the call site as well as information about the global variable Scanner whose definition is several hundred lines away from the call on line 4 in the snippet.

### 3.1 JSelect

Our statement selection algorithm, JSelect, is an intra-procedural def-use analysis combined with a simple window around the call site. We show an example of our statement selection in Appendix B in Figure 6 and Table 7 with other statement selection methods we tested.

Typically, static program slicing (Weiser, 1984) would be used to identify the relevant statements to the function call. The goal of static slicing is to identify all statements that can affect a specified variable, and has been used with machine learning algorithms to perform type inference for Python (Yan et al., 2023). Unfortunately, static program slicing for Javascript runs into the same problem as call graph reconstruction, as it requires a pointer analysis to collect the data flow graph. Sridharan *et al.* showed that a traditional field-sensitive pointer analysis has an $O(N^4)$ runtime for an N-statement Javascript program (Sridharan et al., 2012). In Section 4.1, we show that a field-sensitive pointer analysis is unable to finish on large Javascript libraries within 12 hours. To the best of our knowledge, there are no static slicing algorithms using field-sensitive pointer analysis for Javascript that scale to large frameworks like React. Instead, we opt for a cheap and scalable analysis in JSelect to identify the relevant statements for the call sites.

**Call site statement selection.** The intuition behind our approach is to collect the immediate context in which the call is invoked, as well as to capture references to global objects. For example, in Figure 4, we want to capture that the call on line 4 is happening inside of an `if` statement which is inside of a loop. However, we also want to provide information about the global variable `Scanner`.

The implementation of JSelect relies on parsing Javascript AST's to extract variable information. We build our implementation using Esprima (esp) and Esrefactor (Ariya), two static analysis tools for parsing Javascript AST's. First, JSelect performs a static scope analysis in order to each variable to its set of references and declaration. JSelect takes in a call site location and identifies what variables are being referenced in the call itself (in Figure 4, the variables are `Scanner.attributes`, `attr`, `options`, `el`). JSelect collects each of the statements that modifies or uses one of the variables. The final output of JSelect which is sent to the LM contains all of these statements along with the statements immediately surrounding the call site. This approach combines the local context with information about any global variables that are being referenced. JSelect is not path or flow-sensitive, and is not inter-procedural. An inter-procedural analysis would require a pre-existing call graph, leading to a chicken-and-egg problem.

**Target function statement selection.** There is often useful information around the function definition for the target function (If we are trying to determine whether call site `A` calls function `B`, the target function is `B`). For example, the target function may be assigned as a field to an object. In this case, it would be very helpful to include the entire object in the input to the LM. To obtain the relevant information, we use Esprima (esp) to obtain the AST node corresponding to the target function. We then provide the statements of the parent of the target function node to the model. If the parent node is too big (sometimes the parent node is the entire program), we use a window of 50 lines before and after the target function. We set the limit of the parent node at 200 lines.

## 3.2 Prompt Formulation

Once we have filtered the relevant statements from the programs, we form the input to the language model. We base our prompting strategy off of how a human might resolve an indirect call. Our prompting has three steps: program understanding, call site intention inference, and final prediction. Our final prompting template can be seen in Figure 5 in Appendix A.

**Program understanding.** First, the model should gain an understanding of what each statement in the program is doing. We ask the model to interpret the slice from JSelect line-by-line to get a general understanding of the code. Section 4.2 shows the performance of CALLME with different methods of interpreting the code line-by-line, such as asking the model to explain each line or asking the model to simulate the execution line-by-line.

**Call site intention inference.** Next, we ask the model to reason about the call site itself. We provide the call, the line number, as well as information on the variables referenced in the call site. This provides additional information about what the call is being used for and helps determine whether it is likely to be a match with the target function provided later.

**Final prediction.** Finally, we ask the model to predict whether the call site in question refers to the specified function. By this point, the model has reasoned about the program snippet as a whole as well as how the call site of interest is used.

## 4 Evaluation

Our primary baseline is Jelly, an open-source call graph construction algorithm written by the authors of JAM (Nielsen et al., 2021a), which handles dynamic property accesses with string prefixes and postfixes as well as indirect calls. The static analysis for Jelly is based off of three static analysis tools for Javascript – JAM (Nielsen et al., 2021a), ACG (Feldthaus et al., 2013), and Tapir (Møller et al., 2020a). First, we analyze how well CALLME performs in resolving dynamic property calls. Next, we explore how the design decisions in CALLME affect the performance. Finally, we show how CALLME can be used with downstream program analysis tasks such as bug detection. We conducted the experiments on a Linux server with two AMD EPYC 7763 64-Core Processors, 128 cores, 1024GB RAM, and 4 NVIDIA RTX 6000 Ada Generation GPUs.

**Dataset.** We use the dataset from Chakraborty et al. (2022) root cause analysis of Javascript call graphs (Chakraborty et al., 2022), consisting of caller-callee pairs generated by performing dynamic analysis on the popular TodoMVC Suite (Chakraborty et al., 2022). We identified 660 caller-callee pairs that were missed by ACG due to a dynamic property access. For each caller-callee pair, we generate a negative example by selecting a random function for the same caller for a total of 1,320 total caller-callee pairs evenly split between positive and negative samples. Statistics for each framework in the TodoMVC Suite can be found in Table 1.

| Program | #Lines | Inter-File | Total |
|---|---|---|---|
| AngularJS | 12,091 | 41 | 207 |
| Backbone | 9,003 | 22 | 46 |
| KnockoutJs | 1,044 | 5 | 24 |
| KnockbackJs | 15,836 | 24 | 85 |
| CanJs | 11,371 | 34 | 91 |
| React | 24,855 | 11 | 57 |
| Mithril | 1,433 | 25 | 27 |
| Vue | 7,667 | 15 | 61 |
| VanillaJs | 751 | 0 | 14 |
| jQuery | 9,526 | 7 | 48 |
| **Total** | **93,557** | **184** | **660** |

Table 1: Dataset statistics. **Inter-file** refers to the number of caller-callee edges that are in different files, and **Total** refers to the total number of calls.

### 4.1 Performance on TodoMVC

Table 2 presents the results of CALLME while using different LM backends. We test the CodeLlama models (Rozière et al., 2024b), Llama-3.3-70B-Instruct (Grattafiori et al., 2024), and GPT-4 (Achiam et al., 2023). CALLME with CodeLlama-34b achieves an F1 score of .79 and is able to detect almost 7x more calls as Jelly while maintaining a tolerable false positive rate. Additionally, as shown in Table 9 in Appendix D, 39/71 of the

| Model | Detect | Miss | FP | Acc. | F1 | Prec. | Recall |
|---|---|---|---|---|---|---|---|
| Jelly | 71 | 589 | 19 | .501 | .19 | .79 | .11 |
| CodeLlama-7B | 490 | 170 | 339 | .61 | .71 | .59 | .74 |
| CodeLlama-13B | 435 | 225 | 230 | .66 | .66 | .65 | .66 |
| CodeLlama-34B | 497 | 163 | 102 | .80 | .79 | .83 | .79 |
| Llama-3.3-70B | 280 | 380 | 18 | .70 | .59 | .94 | .42 |
| GPT-4 | 292 | 368 | 13 | .71 | .61 | .96 | .44 |

Table 2: Final results on the TodoMVC benchmark using different models for the final inference. FP stands for False Positive.

calls detected by Jelly are in Knockback.js, due to Knockback using more dynamic property accesses with string concatenations. Excluding Knockbackjs, Jelly can only detect 5.5% of calls, while CALLME is able to detect 75% of all calls in our dataset. Additionally, as shown in Table 8 in Appendix D, results remain stable even when calls are resolved across different files. GPT-4 and Llama-3.3's false positive rates are much lower than everything else, with almost

9x fewer false positives than the next lowest model, but suffers from lower recall. CALLME is configurable with different LM backends, so users can use CodeLlama-34b or GPT-4 based on their precision requirements.

**Runtime**. As noted in prior work (Sridharan et al., 2012), a field-sensitive pointer analysis is intractably slow for large programs. To test, we ran a pointer analysis using two existing static tools on the 5 largest files in the TodoMVC benchmark. WALA performs a standard Andersen's alias analysis with call site abstractions. TAJS is a static analysis tool based on abstract interpretation. TAJS errored out on all of the programs except for React due to unsupported Javascript features. We set the time-out threshold at 12 hours.

As CALLME is meant to augment existing static analyzers, we use an analogous setup for our experiment. We run Jelly on each program in Table 3 and find all calls that do not have a target function. We then run CALLME once for each call. This is likely an over-approximation of CALLME's runtime in practice, as we do not anticipate CALLME being used to construct an entire call graph. However, as seen in Table 3, CALLME is still able to scale to much larger programs than a traditional field-sensitive pointer analysis.

| Program | #Lines | #Calls | Total Runtime (Hours) | | |
|---|---|---|---|---|---|
| | | | CALLME | WALA | TAJS |
| AngularJS | 28,363 | 1,349 | 2.6 | T.O (12+) | N/A |
| React | 21,641 | 2,307 | 4.5 | T.O (12+) | T.O (12+) |
| Jquery | 9,205 | 728 | 1.4 | T.O (12+) | N/A |
| Ractive | 9,133 | 900 | 1.7 | T.O (12+) | N/A |
| Blocks | 14,724 | 1,031 | 2 | T.O (12+) | N/A |
| **Average** | 16,613 | 1,263 | 2.5 | N/A | N/A |

Table 3: Runtime performance of CALLME compared to a field-sensitive pointer analysis. TAJS errored out on everything except for React. #Calls refers to the number of call sites which Jelly does not return any target functions.

## 4.2 Ablations

We tested multiple statement selection methodologies and prompting formats to achieve the best tradeoff between scalability and accuracy. We perform the same experiment as in Section 4.1 on the TodoMVC dataset.

**Statement selection.** We evaluate the effect of different statement selection methodologies in CALLME's performance. Our goal is to find the fastest analysis that still gets robust performance. We show an example program in Figure 6, as well as the output of various slicers in Table 7. We test using a simple window around the caller and callee, a program slicer tracking the flow of function values (Feldthaus et al., 2013), thin slicing (Sridharan et al., 2007), and following def-use information of variables used in the call sites. Additional information on each of the statement selection methodologies can be found in Appendix B.

| Slicing Method | Detect | Miss | FP | Improve (+/-) | | |
|---|---|---|---|---|---|---|
| | | | | Detect | Miss | FP |
| Def-Use + Window | **497** | **163** | **102** | 0% | 0% | 0% |
| Window Only | 531 | 129 | 299 | +7.1% | -25.6% | +193.1% |
| Def-Use Only | 491 | 169 | 195 | -1.0% | +4.3% | +91.1% |
| Thin-Slicing | 452 | 208 | 170 | -9.7% | +27.8% | 66.7% |
| Full-Slicing | 438 | 222 | 163 | -13.2% | +36.4% | +59.8% |

Table 4: Ablation on various slicing methods. We treat the first row as the baseline and then compute the improvement of other slicing methods. Green = Performance improvement. Red = Performance decrease. Using a window slightly improves the detection and miss rates, but greatly increases the number of false positives.

The results of the slicing ablation study can be found in Table 4, where we find that combining Def-Use chains and a simple window performs best. The improvement seems to come primarily in the false negative and false positive rates, where the next best slicing method still has 60% more false positives. We find evidence that incorporating static analysis leads to

dramatic improvements over only using LMs. Simply using the 50 statements before and after the call sites almost triple the false positive rate. This is likely due to the fact that simply using a window includes the context in which a call is made, but likely does not include information about the relevant variables used in the call itself.

**Prompting.** We tested multiple prompting strategies, such as asking the model to perform abstract interpretation, asking the model to analyze the code in English, and directly prompting the model. Details on all our prompting strategies can be found in Appendix C.

The results of the study can be found in Table 5, where we find that asking the model to perform abstract interpretation has the best results. We can see that giving the model instructions as well as generating a high level summary through abstract interpretation or English interpretation of the code significantly improves performance over the *direct* and *explain* approaches.

Interestingly, abstract interpretation performs better than English. This is likely due to it being slightly more granular and providing better information about the individual variables.

### 4.3 Bug detection.

In order to determine CALLME's effectiveness in a downstream program analysis task, we use CALLME to help identify bugs which rely on resolving dynamic property accesses. We simulate a taint analysis scenario, where the analysis needs to

| Prompt Method | Detect | Miss | FP | Improve (+/-) | | |
|---|---|---|---|---|---|---|
| | | | | Detect | Miss | FP |
| Abstract Interpret. | **495** | **162** | **102** | 0% | 0% | 0% |
| English Interpret. | 450 | 209 | 94 | -10% | +29% | -8.5% |
| Instruct | 490 | 169 | 165 | -1.0% | +4.3% | +59% |
| Two-Step | 304 | 171 | 119 | -62.8% | +5.6% | +16.7% |
| Direct | 525 | 135 | 277 | +6.1% | -20% | +171.6% |
| Explain | 471 | 189 | 225 | -5.1% | +16.7% | +120.1% |

Table 5: Ablation on various prompts. We treat the first row of each design as the baseline and compute the improvement of other alternatives. Green = Performance improvement. Red = Performance decrease.

discover all paths from any tainted sources to vulnerable sinks. For example, a user might want to locate all call sites to library function which writes information to the filesystem to ensure that it is free from unsafe user input. An example of a real-world security vulnerability that requires resolving a dynamic property call can be found in Appendix E in Figure 7.

Our dataset includes multiple examples of security bugs that require resolving dynamic property access calls such as Cross Site Scripting, Arbitrary File Overwrites, Prototype Pollution, Improper Access Control, and Remote Code Execution.

**Case study methodology.** We manually searched through several hundred examples from three datasets of known Javascript bugs, SecBench.js, BugAid, and Vulnerable Functions in the Wild for bugs which required resolving dynamic property accesses and found 24 separate bugs. For each, we manually identified the target buggy function in question. Next, we built an automated detection tool which performs the following steps:

1. Scans the code for dynamic property access calls with Esprima (esp) and EsRefactor (Ariya), which returns all the locations in the program where dynamic property access calls occur.

2. For each call discovered by our scanner, ran CALLME to obtain the relevant statements to the call site.

3. Formulates the statements into a prompt as described in Section 3 with the vulnerable function to determine whether the call could refer to the vulnerable function or not and query CodeLlama-34b.

Our final dataset has 25 dynamic property access calls which resolve to vulnerable functions and 66 which do not.

**Code obfuscation.** Language models trained on code have been shown to be brittle to semantics-preserving program transformations (Miceli-Barone et al., 2023; Zeng et al., 2022). To test our CALLME's robustness, we obfuscate our samples using UglifyJS (Mishoo) and repeat our experiment. We replace all variable names with single letters, such as a and b, and compress the AST with UglifyJS's built-in compressor. It is important to note that analyzing obfuscated code is a *significantly* harder task. Prior work shows that language models are very brittle to name changes (Miceli-Barone et al., 2023), as replacing random variable names causes *an average of 81% performance decrease in BLEU-4 on code summarization (Zeng et al., 2022)*.

**Case study results.** As shown in Table 6, CALLME successfully resolves 24 true positives with only three false positives and a single false negative. Closer introspection into the failures show that the false negative uses function

| Prediction | Normal | | Obfuscated | |
|---|---|---|---|---|
| | True Pos. | True Neg. | True Pos. | True Neg. |
| Pred. True | 24 | 3 | 17 | 9 |
| Pred. False | 1 | 63 | 8 | 57 |

Table 6: Results of case study on obfuscated and un-obfuscated samples.

parameters as part of the indirect call, which are not handled by JSelect as discussed in Section 3. Additionally, performance remains relatively robust on an obfuscated dataset, where CALLME can still identify almost 67% of true positives.

# 5 Related Work

**Program analysis for Javascript.** There have been several approaches to static analysis for Javascript (Lee et al., 2012; Jensen et al., 2009; Sridharan et al., 2012; Møller et al., 2020b; Nielsen et al., 2021b; Li et al., 2022; Kang et al., 2023). Several tools also explicitly target call graph construction (Feldthaus et al., 2013; Nielsen et al., 2021a; Madsen et al., 2015). Additionally, these systems have difficulty scaling up to large programs, as shown in Section 4.1. Madsen *et al.* generates a call graph using static analysis, but only handles event listeners. ACG (Feldthaus et al., 2013) and JAM (Nielsen et al., 2021a) generate call graphs for Javascript programs that can handle multiple files and are scalable, but ignore dynamic property accesses.

There have been several analysis techniques to apply additional analysis specifically for certain features such as dynamic reads and writes (Park et al., 2013; Ko et al., 2019; Stein et al., 2019; Kim et al., 2014; Madsen & Andreasen, 2014; Park et al., 2016). One popular technique is to use dynamic information to focus a static analyzer on dynamic structures collected at runtime, and has led to many improvements in reasoning about dynamic data (Wei et al., 2016b; Wei & Ryder, 2014b; Wei et al., 2016a; Wei & Ryder, 2013; 2014a; 2012; Wei, 2012; Chakraborty et al., 2024; Laursen et al., 2024). However, these approaches are not purely static like CALLME, as they require the program to be executed to collect the runtime information.

**LMs for program analysis.** LMs have been used for many program analysis task such as type inference (Peng et al., 2023; Wei et al., 2023; Wang et al., 2023b), fuzzing (Xia et al., 2024; Yang et al., 2023b;a; Deng et al., 2023), vulnerability detection (Mathews et al., 2024; Liu et al., 2023), resource leak detection (Wang et al., 2023a; Mohajer et al., 2023), code summarization (Cai et al., 2023b; Geng et al., 2024; Ahmed et al., 2024; Wang et al., 2022), and fault localisation (Wu et al., 2023). DLInfer (Yan et al., 2023) and TypeGen (Peng et al., 2023) both use static program slicing as well as a machine learning model for type inference on Python. CALLME differs from the prior work as it is the first to perform call graph construction. Unlike many prior works, CALLME is meant to augment current static analysis tools on certain edge cases, rather than replacing them. To the best of our knowledge, CALLME is the first work to explicitly tackle an inter-procedural analysis task like call graph construction using LMs.

## 6    Limitations

**Single link prediction.** CALLME is designed to predict a single edge between a specified call site and a function. Thus, it is not scalable to build an entire call graph using CALLME, and is not designed for a task such as IDE support like ACG (Feldthaus et al., 2013) on its own. However, as many program analysis tasks only require discovering a single link, CALLME can still be useful in many cases as demonstrated in Section 4.3. Additionally, CALLME is meant to be used alongside existing algorithms to handle some difficult edge cases rather than replacing them.

## 7    Conclusion

We present a new approach, CALLME, which combined static program analysis with LLMs to perform call graph construction for Javascript. Given a full program, CALLME first runs the static analysis, JSelect, to select the relevant statements. Next, CALLME formulates a prompt with the output of JSelect and queries an LLM to determine whether a specified call site calls a target function. We evaluate CALLME on the TodoMVC benchmark (Chakraborty et al., 2022) and find that CALLME is able to resolve 75% of all calls with an F1 score of .79. The next best tool, Jelly, is only able to resolve 11% of these calls with an F1 score of .19. Additionally, we show how CALLME can find vulnerabilities that are currently undetectable by static analyzers, where we are able to find 24 out of 25 vulnerabilities with only 3 false positives.

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

```
    [CALLER SNIPPET from JSelect]
Please perform abstract interpretation on the previous piece of code line by line.
After each line you interpret, provide the output from abstract interpretation.
Focus specifically on the call `[CALL]` that is on line [LINE]. Please note that
this is not a complete piece of code, and is a snippet from a larger body.

              [MODEL RESPONSE]

What does the `[CALL]` call on line [LINE] do? Below is some information about the
variables that are used.
Var1, which is defined on line X of the previous snippet.
Var2, which is defined on line Y of the previous snippet.
...
              [MODEL RESPONSE]

Does the `[CALL]` call refer to the function `[TARGET FUNCTION]` on line [LINE] in
the following program snippet? Please use the information provided previously and
answer yes or no.
    [TARGET FUNCTION SNIPPET from JSelect]

              [MODEL RESPONSE]
```

Figure 5: The final prompting template for CALLME.

```
1   var obj = {};
2   var callUserString = true;
3   obj["call_func1"] = function() {
4       console.log("func1");
5   }
6
7   obj["call_func2"] = function() {
8       console.log("func2");
9   }
10
11  function callFunc(userString) {
12      if (userString == "func1" ||
13          userString == "func2") {
14          if (callUserString === true) {
15              var a = "call_" + userString;
16              obj[a]();
17          }
18      } else {
19          obj[userString] = "foo";
20      }
21  };
22
23  function foo() {
24      callFunc("func1");
25  }
26  foo();
```

Seed statement

Figure 6: An example of a program to demonstrate slicing.

# A  Appendix

# B  Statement Selection Details

*Simple Windowing*: Our baseline approach is to use a simple window around the call site and callee function without any further analysis. This determines whether LLMs can perform call graph reconstruction without being augmented by traditional static analysis tools. In our experiments, we found a window size of 50 to work best.

*Full Slicing with ACG*: As CALLME is meant to be used alongside existing algorithms such as ACG, we use the output of WALA's ACG implementation to construct our slices. We

Table 7: The output of different slicing methods on Figure 6.

| Slicing Option | Slice |
|---|---|
| Full Slicing | ```javascript
var obj = {};
obj["call_func1"] = function() {
  console.log("func1");
}
obj["call_func2"] = function() {
  console.log("func2");
}
function callFunc(userString) {
  if (userString == "func1" ||
    userString == "func2") {
    if (callUserString === true) {
      var a = "call_" + userString;
      obj[a]();
    }
  }
}
callFunc("func1");
``` |
| Thin Slicing | ```javascript
var obj = {};
function callFunc(userString) {
  a = "call_" + userString;
  obj[a]();
}
callFunc("func1");
``` |
| Def-Use | ```javascript
var obj = {};
obj["call_func1"] = function() {
    console.log("func1");
}

obj["call_func2"] = function() {
    console.log("func2");
}
var a = "call_" + userString;
obj[a]();
obj[userString] = "foo";
``` |

choose ACG rather than Jelly as its implementation inside of WALA allows for querying of the points-to graph generated during call graph construction. ACG generates a simple flow analysis (Feldthaus et al., 2013), which only tracks the flow of function values rather than all objects. This allows it to scale to large frameworks such as *React* and *jQuery*. We use the output from the flow analysis to build a system dependence graph for our slicer. WALA constructs an Intermediate Representation (IR) from the source code. We use the built in source mapping to find all IR statements that correspond with the caller line. Each statement is a seed statement. Full slicing returns the set of all statements that can influence the value of one or more specified seed statements by tracing all data and control dependencies from any variables in the seed statement, and is repeated recursively to find all statements. Table 7 shows the output of our full slicer on the code from Figure 6. Note that ACG only tracks function values rather than all objects, leading to the definition of `callUserString` on line 2 not being included in the slice.

*Thin Slicing with ACG*: Full program slices often return more information than is necessary. To remedy this, Sridharan *et al.* developed an approach called thin slicing (Sridharan et al., 2007), which helps limit the size of the slices. Thin slicing ignores value flow of base pointers as well as control dependencies. In contrast, if one of the seed statements contains a value read from a global object, a full slice would include *all* writes to that object, whereas a thin slice would ignore all writes to the object. In Table 7, we can see that the thin slice ignores all control dependencies and all other writes to `obj`.

| Model | Detect | Miss | FP | Acc. | Prec. | Recall |
|---|---|---|---|---|---|---|
| Jelly | 7 | 177 | 0 | 52% | 1.0 | .04 |
| CodeLlama-7B | 108 | 76 | 49 | 66% | .69 | .59 |
| CodeLlama-13B | 133 | 51 | 35 | 76% | .79 | .72 |
| CodeLlama-34B | 121 | 63 | 21 | 77% | .85 | .66 |
| GPT-4 | 98 | 86 | 3 | 76% | .97 | .53 |

Table 8: Results to determine CALLME's ability to resolve calls from different files.

*Def-Use.* We follow the protocol described in Section 3. The Def-Use slice only contains statements that reference the one of the variables at the seed statement. In the code snippet in Figure 6, these are any statements that refer obj or a. We test both using only the Def-Use information as well as combining it with a naive window, as seen in Table 4.

## C   Prompting details.

*Program understanding with Abstract Interpretation.* The approach described in Section 3. We ask the model to perform abstract interpretation line-by-line in the first prompt. We then ask the model to analyze the purpose of the call. Finally, we ask the model whether the call refers to the target function.

*Program understanding with English Interpretation.* This serves as a baseline to determine how much asking the model to perform abstract interpretation helps. We use the same prompts as the abstract interpretation approach, except we ask the model to create textual descriptions of each line without mentioning abstract interpretation.

*Instruct.* We guide the model step by step through resolving a dynamic property access call using the same methodology that a person would. The prompts in order are as follows:

1. We ask the model to analyze the variables used in the call, and provide the variable information that is provided by JSelect as well as the statements.

2. We ask the model what the purpose of the call is for and how it is used.

3. We ask the model whether the call refers to the target function and provide the statements for the target function output by JSelect.

*Two-step.* We first provide the model with the relevant statements for the call site, and ask the model to analyze what is happening at the call site. After the model responds, we then provide the model with the callee function and ask whether the call site refers to the callee function. The purpose of the two-step prompt is to see if the model is able to determine what the important pieces of information are in resolving a dynamic property access.

*Direct.* In our direct prediction approach, we provide the model with the relevant statements for the call site as well as the callee function, and ask the model to predict whether the call site can refer to the callee function.

*Explain.* We perform the same test as in the direct prompts, but also ask the model to explain its reasoning before providing an answer.

## D   Additional Results

## E   Vulnerability Example

**Motivating example**.  To show how resolving dynamic property accesses can be crucial to vulnerability detection, consider the following code snippet from the NPM package find-process in Figure 7, which has over 1.2 million weekly downloads (fin).

| Benchmarks | Detect | | Miss | | FP | |
|---|---|---|---|---|---|---|
| | CALLME | Jelly | CALLME | Jelly | CALLME | Jelly |
| AngularJS | 136 | 12 | 71 | 195 | 25 | 11 |
| Backbone | 30 | 2 | 16 | 44 | 4 | 1 |
| KnockoutJs | 24 | 2 | 0 | 22 | 6 | 0 |
| KnockbackJs | 70 | 39 | 15 | 46 | 15 | 4 |
| CanJs | 71 | 6 | 20 | 85 | 18 | 2 |
| React | 45 | 2 | 12 | 55 | 15 | 0 |
| Mithril | 27 | 1 | 0 | 26 | 4 | 0 |
| Vue | 48 | 4 | 13 | 57 | 3 | 1 |
| VanillaJs | 13 | 0 | 1 | 14 | 2 | 0 |
| jQuery | 32 | 3 | 16 | 45 | 10 | 0 |

Table 9: Results separated by program and compared to Jelly, the state of the art call graph construction algorithm for Javascript.

```
1   const finders = {
2     darwin : function( cond ) {
3       ...
4       ...
5       ...
6       exec(cond.cmd);              Command injection!
7     },
8     android: function(cond) {
9       let cmd = 'ps';
10      utils.exec(cmd);
11    },
12    'linux': darwin
13  }
14
15  function findProcess (cond) {
16    let platform = process.platform;
17    let  find = finders[platform];
18    if (typeof find === 'string') {
19      find = finders[find];
20    }
21    find(cond) .then(resolve, reject);
22  }
23
24  module.exports = findProcess
```

Figure 7: A security vulnerability from findProcess. Because cond is an argument to the darwin function, whether there is command injection on line 6 depends on whether cond.pid was sanitized before the call to darwin. Resolving the call on line 21 requires resolving the dynamic property access on line 17.

There is a potential command injection vulnerability in the function darwin on line 6. If an attacker can control cond.pid, they can execute arbitrary commands as cond.pid is passed into exec. However, cond is a function argument, so whether or not there is a vulnerability depends on whether it was sanitized before the function call. The function call is made on line 21, where if finders[platform] is darwin, then the darwin function is called with unsanitized input. In order to discover the vulnerability, a static analyzer would need to determine whether the dynamic property access on line 17 can refer to the darwin function on line 2. As a result, current static call graph algorithms such as JAM (Nielsen et al., 2021a) and ACG (Feldthaus et al., 2013) miss this dynamic property access and the vulnerability is undiscovered.

