# OpenReview forum: "CALLME: Call Graph Augmentation with Large Language Models for Javascript"
_colmweb.org/COLM/2025/Conference — COLM 2025_

### Official Review · Reviewer_7TUv · 2025-05-05

**Rating:** 6
**Confidence:** 3
**Ethics Flag:** 1

**Summary:**

This paper explores the use of LLMs for call graph analysis for Javascript programs, specifically for resolving dynamic calls that are beyond the scope of static analysis. Experimental evaluation shows that a combination of statement selection and LLM prompting significantly improves the recall on dynamic calls compared to a baseline system using traditional static analysis. The implemented system CaLLME is apparently the first static analysis system capable of resolving dynamic property access without runtime information.

**Questions To Authors:**

It is not clear to me how challenging the evaluation data sets are. If I understand correctly, the TodoMVC data set originally consists of positive examples of caller-callee pairs, and has been augmented with negative pairs constructed by pairing a caller in a positive pair with a random function. This raises the question of whether the negative examples are at all plausible, which may undermine the reliability of the evaluation. Similarly, the case study reported in Section 4.3 seems to be based on a small number of hand-picked examples, which makes it hard to assess how challenging the evaluation is, especially in the absence of a baseline.

Another aspect of the evaluation is the choice of baselines. for the TodoMVC evaluation, the baseline is Jelly, which is described in the opening paragraph of Section 4, but there is no information about how this system was selected, nor what other systems could have been chosen. And, as already noted, in the case study in Section 4.3, there is no baseline at all.

In my view, the narrative style in Section 4 can be improved by clearly separating experimental setup, results, and analysis, instead of presenting everything in one narrative flow, although I am aware that this partly a matter of individual taste.

I also find the first argument in favor of using LLMs (on page 2) a bit strange. Essentially, what the authors say is that, since all existing methods, are unsound and incomplete, it is okay to use another unsound and incomplete method. This may be true, but it is hardly a strength of the proposed method.

Typos and such:
- Page 2: applied downstream -> applied to downstream
- Page 3: Under "Arity mismatching", there is a reference to line 8 in Figure 2, which is a blank line.
- Page 7, Table 4: There is a missing plus (or minus) sign in the last column of the "Thin-Slicing" row.

**Reasons To Accept:**

- Novel approach to call graph analysis, as well as a novel task for LLMs.
- Promising experimental results with some analysis.

**Reasons To Reject:**

- There is no obvious reason to reject, but there are aspects of the evaluation that at least requires some clarification.
- The presentation can be made clearer.

---

> ### Author Response · Authors · 2025-06-03
> **Response to weak evaluation datasets and baselines**
>
> Thank you for your comments! We have added our responses to your questions below.
>
> > It is not clear to me how challenging the evaluation data sets are. If I understand correctly, the TodoMVC data set originally consists of positive examples of caller-callee pairs, and has been augmented with negative pairs constructed by pairing a caller in a positive pair with a random function. This raises the question of whether the negative examples are at all plausible, which may undermine the reliability of the evaluation. Similarly, the case study reported in Section 4.3 seems to be based on a small number of hand-picked examples, which makes it hard to assess how challenging the evaluation is, especially in the absence of a baseline.
>
> **TodoMVC difficulty.** We would like to clarify that this setup of using randomly sampled negative edges reflects our intended binary decision task. CALLME is designed for targeted use cases (e.g., taint tracking, security auditing) where the goal is to determine whether a specific edge exists between a dynamic call site and a known function of interest. In such scenarios, negative examples are naturally defined as arbitrary functions that are not reachable from the call site. This framing matches real downstream usage, and we’ll make this clearer in the revised paper.
>
> That said, we agree that exploring *hard negatives*—e.g., semantically similar or lexically confusable functions—would be an interesting direction for future work and could help probe model robustness.
>
> **Case Study.** For the case study in Section 4.3, the examples are not arbitrarily hand-picked; they are systematically curated from public vulnerability benchmarks (SecBench.js, BugAid, and Vulnerable JavaScript Functions in the Wild) and filtered to include only those requiring dynamic property resolution. We did not include a baseline in the paper because static analyses don’t support dynamic property resolution. We have run a separate experiment showing that Jelly is unable to resolve any of the edges shown in the table below. CALLME can resolve 96% with only 3 false positives (Table 6 in the paper). We will clarify this point in the paper and include a summary table in the appendix to improve transparency.
>
> |                    | True Positive | True Negative |
> | ------------------ | ------------- | ------------- |
> | Predicted Positive | 0             | 0             |
> | Predicted Negative | 24            | 67            |
>
> ---
> > Another aspect of the evaluation is the choice of baselines. for the TodoMVC evaluation, the baseline is Jelly, which is described in the opening paragraph of Section 4, but there is no information about how this system was selected, nor what other systems could have been chosen. And, as already noted, in the case study in Section 4.3, there is no baseline at all.
>
> We selected Jelly as the primary baseline because it is the most precise and actively maintained static call graph construction tool for JavaScript, and was specifically introduced to improve upon older tools like ACG. We will clarify this in the paper.
>
> **We also evaluated two other widely cited static analyzers**—WALA and TAJS—but both tools timed out on all TodoMVC examples, failing to generate usable call graphs (Table 3). These results are consistent with prior work noting scalability and precision limitations in these systems, especially in the presence of dynamic property accesses.

---

> > ### Author Response · Authors · 2025-06-03
> > **Response to narrative style and unsoundness**
> >
> > > In my view, the narrative style in Section 4 can be improved by clearly separating experimental setup, results, and analysis, instead of presenting everything in one narrative flow, although I am aware that this is partly a matter of individual taste.
> >
> > We appreciate the feedback on the narrative flow in Section 4 and agree that separating experimental setup, results, and analysis more explicitly could improve clarity for some readers. We will revise this section to better structure the presentation of our experiments, with clearer transitions to help readers distinguish the components of each evaluation.
> >
> > > I also find the first argument in favor of using LLMs (on page 2) a bit strange. Essentially, what the authors say is that, since all existing methods, are unsound and incomplete, it is okay to use another unsound and incomplete method. This may be true, but it is hardly a strength of the proposed method.
> >
> > We understand and agree that unsoundness is not inherently a strength. Our intent was to clarify, not justify, that the class of dynamic calls we target (e.g., `obj[f]` with unresolved `f`) is *already beyond the scope of sound static analysis*. Existing tools skip these cases entirely.
> >
> > In static analysis, using LLMs often raises concerns about compromising soundness. We emphasized this point to clarify that *CALLME does not worsen any existing guarantees*—it operates in a region where no sound method currently applies. We’ll revise the paper to make this motivation clearer and avoid implying that unsoundness is a desirable property

---

> > > ### Comment · Reviewer_7TUv · 2025-06-04
> > > **Thanks**
> > >
> > > Thanks for the clarifications regarding the evaluation.

---

### Official Review · Reviewer_vcVB · 2025-05-14

**Rating:** 4
**Confidence:** 3
**Ethics Flag:** 1

**Summary:**

This paper introduces CALLME that combines the Large Language Model with the static analysis tools to resolve dynamic property calls in Javascript. Dynamic property calls are tough to resolve using static analysis. CALLME first uses the JSelect algorithm to select the potential relevant statements and then construct the prompt for the LLMs to resolve the dynamic property access.

In the paper, CALLME is applied to the Javascript root cause analysis benchmark with manually crafted caller-callee pairs, and shows superior performance compared with the state-of-art tools. Furthermore, CALLME is shown to be helpful in detecting vulnerabilities.

**Questions To Authors:**

1. For TodoMVC, a random function is used as negative example. If I understand correctly, the task is a binary classification task. Does it align with the actual use case of CALLME where there could be many candidates?

2. What techniques do you use to find the potential target functions? How to precisely find the target functions based on AST node?

**Reasons To Accept:**

1. The research problem is well-motivated and suitable to use LLMs to solve.
2. CALLME shows improved performance in resolving a dynamic property access calls.
3. Ablations clearly demonstrate the effectiveness of different design choices.

**Reasons To Reject:**

1. Overall, I do not think COLM is the best venue for this paper for the following reasons. First, the audience generally has limited background in software analysis, especially for Javascript. Some terms, such as 'inter-procedural analysis task', 'taint analysis', and 'abstract interpretation', need to be explained more clearly in the paper. Second, CALLME is mainly a combination of software tools and large language models, with little innovation on the language model side.

2. The problem addressed in the paper is relatively narrow, focusing only on resolving dynamic calls in Javascript.

3. The evaluation could be improved. The current evaluation does not provide an apple-to-apple comparison. In TodoMVC, CALLME is tested as a binary classification method, while the baseline is used to construct a call graph. This setting does not make sense to directly compare CALLME and Jelly. I recommend that the authors propose a more realistic use case where CALLME is evaluated in combination with Jelly and showing improved constructed call graph. Binary classification seems unreasonable in this particular dataset TodoMVC. Furthermore, authors could consider using CALLME with existing static analysis tools on a real use case or a more complex task, rather than only on a curated subset. For example, showing that CALLME can reduce the error-out rate or runtime of existing tools.

4. The presentation of the paper needs improvement. Many results and technical details, such as statement selection strategies, are placed in the appendix, which makes the main paper difficult to follow. Figure 3 lacks the caption and necessary text to demostrate the workflow of CALLME. I also suggest including at least one example from the TodoMVC dataset to help readers better understand the task.

---

> ### Author Response · Authors · 2025-06-03
> **Response to venue fit at COLM**
>
> Thank you for your comments! We have added our responses below.
>
> > Overall, I think COLM is not the best place for this paper for the following reasons: 1) the audience generally have little knowledge on software analysis especially in Javascript. Some terms need to be explained in the paper, for example 'inter-procedural analysis task', 'taint analysis', 'abstract interpretation' etc. 2) CALLME is the combination of software tools and LLMs with little innovation in the side of LLMs.
>
>
> **(1) Terminology clarity:** We agree that terms like interprocedural analysis and taint analysis may be unfamiliar to some readers. We’ll revise the paper to define these terms inline to ensure clarity for a broader audience.
>
> **(2) LM contribution.** We believe that CaLLME’s contribution lies in demonstrating a concrete and generalizable neuro-symbolic approach: using a language model to extend the capabilities of static analysis in domains with dynamic or underspecified semantics. In CALLME, we target dynamic property accesses in JavaScript (e.g., calls of the form obj[f]) which are explicitly excluded by current static tools. The static analysis defines the candidate space by identifying potential callers and callees, and the LM performs inference, forming a clear division of labor.
>
> More broadly, this structure reflects a growing design space for neuro-symbolic systems in code: use symbolic methods to constrain the problem space and use LMs to resolve components that are too challenging for traditional symbolic methods. We believe this takeaway is relevant beyond JavaScript analysis. Many languages and tasks (e.g., dynamic dispatch, API misuse detection, type prediction) involve elements that are challenging for purely symbolic techniques. CALLME offers a compact, transferable modeling pattern for applying LMs in such settings, and we will revise the paper to make this framing clearer for the COLM audience.
>
> We also note that COLM’s call for papers explicitly welcomes “17. LMs with **tools and code**, integration with tools and APIs, and LM-driven software engineering,” which we believe CALLME is directly aligned with. We will revise the paper to better emphasize this framing and its relevance to the COLM audience.

---

> > ### Author Response · Authors · 2025-06-03
> > **Response to presentation improvements.**
> >
> > > I believe the presentation of the paper needs improvement. Many results and details of the technique are in the appendix, e.g. statement selection strategies which makes the paper hard to read. Additionally, I hope the paper can provide at least one example from the TodoMVC dataset so that the audience will have a better understanding of the task.
> >
> > **On the presentation.** To clarify, the core components of CALLME’s method (including the statement selection component, JSelect) are described in the main body of the paper (Section 3). The appendix is used to report ablation-specific details for all experimental variants, not for defining the main pipeline. That said, we recognize that this structure may have made it harder to distinguish the key contributions from the ablations. In our revision, we will more clearly label which parts of the pipeline are core versus ablation-only, and reorganize the presentation to ensure that the full methodology is self-contained and easy to follow without needing to consult the appendix.
> >
> >
> > **Example from TodoMVC.** We do include a motivating example (Figure 2) that originates from the jQuery library bundled with the TodoMVC dataset. However, we understand that its connection to TodoMVC and its role in illustrating the task may not have been sufficiently clear. Because the resolution requires interprocedural reasoning across library and application code, we shortened the example for space, which may have obscured some context. In the revision, we will clarify the provenance of the example, walk through how the dynamic call is resolved, and explicitly highlight the LM’s prediction task and candidate space to better illustrate the challenge and how CALLME addresses it.

---

> > > ### Author Response · Authors · 2025-06-03
> > > **Response to other questions**
> > >
> > > > For TodoMVC, a random function is used as negative example. If I understand correctly, the task is a binary classification task. Does it align with the actual use case of CALLME where there could be many candidates?
> > >
> > > CALLME is intentionally framed as a binary classification task to reflect its intended usage in targeted, on-demand scenarios. Many downstream tools such as taint tracking, vulnerability auditing, or static security policy enforcement don’t need the full call graph. Instead, they ask focused questions like: does this dynamic call site ever reach a dangerous function like `child_process.exec`? In such cases, the relevant question is whether a given call edge is valid, not selecting among all possible callees. This is where CALLME is most useful, and where symbolic static analysis provides no answers. We will revise the paper to articulate this usage pattern and add a limitations section to distinguish CALLME from full call graph construction approaches.
> > >
> > >
> > > > What techniques do you use to find the potential target functions? How to precisely find the target functions based on AST node?
> > >
> > > CALLME is designed for targeted, context-specific queries rather than exhaustive call graph construction. In these scenarios, the **set of potential target functions is defined by the downstream task**. This aligns with how call graphs are typically used in practice; as **intermediate representations** in downstream analyses where only a small number of unresolved call edges are relevant. Here are examples of how the candidate callees are determined in practice:
> > >
> > > - **Taint analysis:** The candidate function is a known sink (e.g., `child_process.exec`), and the analysis checks whether a specific dynamic call could invoke it.
> > > - **Dead code detection:** The candidate is a function suspected of being unreachable. The goal is to determine whether it could be dynamically invoked.
> > > - **Patch validation or repair:** The candidate is a fix function, and we want to verify whether it could plausibly be invoked by a given dynamic call site.
> > >
> > > Once a downstream task identifies a candidate function of interest (e.g., a known sink, a patch function, or an unused export), we can precisely locate its AST node by matching its declared name, location, or export path. From there, we can construct CALLME queries pairing this function with relevant dynamic call sites. These call sites are already identifiable via the AST (e.g., computed property accesses like `obj[f]`). We will clarify this process in the paper.

---

> > ### Comment · Reviewer_vcVB · 2025-06-08
> >
> > Thanks authors for their response. They are helpful and informative.
> >
> > Two more questions on the evaluation:
> > 1. In section 4.1, when comparing Jelly with CALLME, Jelly is constructing the call graph while CALLME is identifying the callee between two candidates. Is it fair to compare the two?
> > 2.  I got the idea that the candidate target called functions are depend on the downstream tasks. When comparing the runtime in Section 4.1, how are the candidate callees be selected? Again, is it fair to compare CALLME with other baselines? Based on the paper, CALLME only runs on a subset of examples where Jelly is not able to resolve.

---

> > > ### Author Response · Authors · 2025-06-09
> > > **Response to additional questions**
> > >
> > > Thank you for your response! We have responded to your additional questions below.
> > >
> > > > 1. In section 4.1, when comparing Jelly with CALLME, Jelly is constructing the call graph while CALLME is identifying the callee between two candidates. Is it fair to compare the two?
> > >
> > > Thank you for raising this important clarification. __Jelly does perform some rudimentary analysis for dynamic properties__, unlike older systems like ACG which ignore them entirely. This allows us to compare the dynamic property resolution of Jelly to that of CALLME. However, Jelly still fails to resolve many dynamic accesses, due to the inherent imprecision of static string analysis in these cases.
> > >
> > > Importantly, no existing call graph construction algorithm handles these dynamic accesses precisely due to the time complexity; resolving dynamic property accesses is O(n⁴) in program size, making precise static resolution intractable in practice [1]. This is a key motivation for CALLME; it leverages language models to recover edges in cases where purely static reasoning fundamentally breaks down.
> > >
> > > So, while CALLME and Jelly operate at different scales, they address the same subproblem at these unresolved call sites. Our evaluation focuses specifically on those shared cases to demonstrate how LLM guidance can meaningfully extend the coverage and precision of existing static tools.
> > >
> > > > 2. I got the idea that the candidate target called functions are depend on the downstream tasks. When comparing the runtime in Section 4.1, how are the candidate callees be selected? Again, is it fair to compare CALLME with other baselines? Based on the paper, CALLME only runs on a subset of examples where Jelly is not able to resolve.
> > >
> > > Thank you for the thoughtful question — we’re happy to clarify.
> > >
> > > In the runtime experiment in Section 4.1, our goal is to isolate and measure the overhead of invoking CALLME at a call site, independent of any downstream task. Since accuracy is not the focus here, we simply run CALLME __once per unresolved call site__, using a __randomly selected candidate callee__.
> > >
> > > This avoids entangling the runtime with candidate selection strategies, whose runtimes vary depending on downstream use cases. Our intent is to measure __just the cost of CALLME itself__, not any particular application.
> > >
> > > The runtime numbers include both __running Jelly__ and __running CALLME on each unresolved call site__, and are compared to __field-sensitive whole-program static tools__ like TAJS and WALA, which attempt to handle dynamic property accesses statically but time out.
> > >
> > > We believe the comparison is fair because it reflects two realistic options for handling dynamic property accesses:
> > >
> > > 1.  Run a fast static analysis like Jelly, and use CALLME selectively for unresolved cases.
> > > 2.  Attempt to resolve everything statically with a full field-sensitive analysis like TAJS or WALA.
> > >
> > > Our evaluation demonstrates that the first approach is significantly more tractable, while still recovering high-quality information at challenging call sites.
> > >
> > > We will update the next version of the paper by clarifying that the CALLME column includes the time for both Jelly and CALLME in Table 3.
> > >
> > > [1] Sridharan, Manu, Julian Dolby, Satish Chandra, Max Schäfer, and Frank Tip. "Correlation tracking for points-to analysis of JavaScript." In ECOOP 2012–Object-Oriented Programming: 26th European Conference, Beijing, China, June 11-16, 2012. Proceedings 26, pp. 435-458. Springer Berlin Heidelberg, 2012.

---

> > > > ### Comment · Reviewer_vcVB · 2025-06-09
> > > >
> > > > Thank you for the clarification. I understand the motivation of the paper and recognize that CALLME addresses a problem that is challenging for existing static analysis tools. However, I have concerns about modeling the task as a binary classification problem without grounding it in a realistic use case. Specifically, the binary classification setup does not seem appropriate for comparison with Jelly. The main problem would be how to select the negative example in order to properly demonstrate CALLME's performance.
> > > >
> > > > As the authors noted, there are several practical scenarios where CALLME could be applied, such as taint analysis, dead code detection, patch validation, or repair. I suggest that the authors describe how CALLME would be used in these scenarios and how it could be integrated with existing tools and conduct the evaluation under the specific use cases. In its current form, the evaluation setup does not appear realistic or convincing.
> > > >
> > > > In addition, the approach for measuring runtime is not realistic under the current setting, which uses one randomly selected candidate callees. For any of the three downstream tasks, evaluating only one candidate callee is insufficient.
> > > >
> > > > In my opition, redesigning the evaluation would make the paper much more solid and interesting.
> > > >
> > > > Thanks again for the response.

---

### Official Review · Reviewer_NECr · 2025-05-15

**Rating:** 5
**Confidence:** 3
**Ethics Flag:** 1

**Summary:**

This paper proposes CALLME, a hybrid technique that combines LLMs and static analysis for call graph construction of JavaScript programs. CALLME implements a lightweight intra-procedural def-use analysis to select relevant statements and uses LLMs to predict. The evaluation on curated datasets of call edge and bug detection show promising results.

**Reasons To Accept:**

+ First hybrid “LLM + static analysis” approach aimed specifically at resolving dynamic property accesses in JavaScript.

+ Reported significant gains +30% in accuracy and 0.60 in F1 over the best‑published static analyzer on the chosen benchmark and near‑perfect recall in the bug detection study.

+ Potential for broad impact for bug‑finding, refactoring, and IDE tooling with better call graph construction.

**Reasons To Reject:**

- Evaluation datasets are cherry-picked to favor CALLME. More specifically, the call‑edge dataset contains only the 660 edges missed by ACG; so baselines are guaranteed with low recall. On bug detection study, 24 vulnerabilities are manually selected so that require dynamic‑property resolution while other bug classes are ignored. I suggest authors to evaluate on larger benchmarks (such as SecBench.js vulnerabilities).

- Negative edge might be weak that might inflate the precision of the proposed approach.

- No evaluation on complete TodoMVC dynamic traces.

- Some inconsistencies in reporting (different accuracy numbers reported: 80% in abstract, 75% in introduction and conclusion; Detect and Miss in some rows of Table 2&4 do not add to 660...).

---

> ### Author Response · Authors · 2025-06-03
> **Response to cherry-picked evaluation datasets.**
>
> Thank you for your thoughtful comments and your thorough reading of the paper! We have responded to the comments below.
>
> > Evaluation datasets are cherry-picked to favor CALLME. More specifically, the call‑edge dataset contains only the 660 edges missed by ACG; so baselines are guaranteed with low recall. On bug detection study, 24 vulnerabilities are manually selected so that require dynamic‑property resolution while other bug classes are ignored. I suggest authors to evaluate on larger benchmarks (such as SecBench.js vulnerabilities).
>
> **Cherry-picked evaluation.**  We do not intend to over-claim that CALLME is better in all aspects of call graph construction. Our evaluation is **intentionally scoped** to measure CALLME’s performance on a specific, well-known failure point in JavaScript call graph analysis: **dynamic property accesses** (e.g., `obj[f]` where `f` is not statically resolvable). Traditional static tools like ACG explicitly ignore these call sites. CALLME is designed specifically to address this **statically identifiable blind spot**, and our evaluation is aligned with that goal. On direct call analysis, ACG should have 100% accuracy conceptually, but CALLME could suffer from potentially unreliable predictions due to the use of LLMs.
>
> **Bug detection dataset.** To clarify, our vulnerability dataset was systematically curated from three established sources: SecBench.js [1], Vulnerable JavaScript Functions in the Wild [2], and BugAid [3] as mentioned in Section 4.3. We didn’t include detailed descriptions for space purposes.
>
> We manually inspected the samples and selected the 24 that require dynamic property resolution, meaning they are **not analyzable by design** using existing static analysis techniques. This is not cherry-picking but t**argeted benchmarking** against a class of real-world bugs that current methods cannot address. We performed a separate experiment running Jelly, the current state-of-the-art call graph constructor for Javascript, to see how many of the calls it could resolve in our case study, and included the table below. It cannot identify any of the bugs. We will clarify this curation process in the paper and include a table in the appendix listing the selected vulnerabilities, their sources, and the dynamic call site involved.
>
>
> |                    | True Positive | True Negative |
> | ------------------ | ------------- | ------------- |
> | Predicted Positive | 0             | 0             |
> | Predicted Negative | 24            | 67            |
>
>
> [1] Bhuiyan et al. (2023). SecBench.js: An Executable Security Benchmark Suite for Server-Side JavaScript. **ICSE 2023**.
>
> [2] Kluban et al. (2022). On Measuring Vulnerable JavaScript Functions in the Wild. **AsiaCCS 2022.**
>
> [3] Hanam et al. (2016). Discovering bug patterns in JavaScript. **FSE 2016.**

---

> > ### Author Response · Authors · 2025-06-03
> > **Response to weak negative edges and full TodoMVC traces**
> >
> > > Negative edge might be weak that might inflate the precision of the proposed approach.
> >
> > In our setting, CALLME is explicitly framed as a binary decision tool: given a dynamic call site and a candidate callee, should there be an edge? The use of randomly selected functions as negative examples reflects this framing: in real-world downstream tasks (e.g., auditing, taint tracking), analysts are often evaluating whether a particular call edge exists, not ranking among all functions in scope. Our use of randomly selected functions as negatives reflects this binary framing. These are not artificially weak negatives, they are simply other real functions in the program, and in practice, analysts may face similarly broad uncertainty about potential callees.
> >
> > That said, exploring more difficult negative examples (e.g., functions with similar structure or naming) would be valuable experiments. We will clarify this point in the paper.
> >
> > > No evaluation on complete TodoMVC dynamic traces.
> >
> > CALLME is designed to resolve **specific dynamic calls that static analyses cannot handle**, not to reconstruct a full call graph. Evaluating against complete dynamic traces would include many trivial edges (e.g., direct calls, statically known properties) that are already resolved by baseline tools and not relevant to CALLME’s purpose. Including those edges would dilute the evaluation, since CALLME was never intended to compete on that ground. Our goal is to evaluate its value in filling a **precise, well-known blind spot** in static analysis: dynamic property accesses that are intentionally ignored by tools like ACG. We’ll clarify this framing more explicitly in the revision.

---

> > > ### Author Response · Authors · 2025-06-03
> > > **Response to reporting inconsistencies.**
> > >
> > > > Some inconsistencies in reporting (different accuracy numbers reported: 80% in abstract, 75% in introduction and conclusion; Detect and Miss in some rows of Table 2&4 do not add to 660...).
> > >
> > > The 75% in the introduction refers to the percentage of calls resolved, not the accuracy – CodeLlama-34B resolved 496/660 (75%) calls. We apologize for the confusion.  We will update this to reflect the 80% in the abstract.
> > >
> > > The inconsistencies in Tables 2&4 were due to a small error in our script. We have fixed it. The updated tables are below.
> > >
> > > | Model         | Detect | Miss | FP  | ACC  | F1  | Prec. | Recall |
> > > | ------------- | ------ | ---- | --- | ---- | --- | ----- | ------ |
> > > | Jelly         | 71     | 589  | 19  | .501 | .19 | .79   | .11    |
> > > | Codellama-7b  | 490    | 170  | 339 | .61  | .66 | .59   | .74    |
> > > | CodeLlama-13b | 435    | 225  | 230 | .66  | .66 | .65   | .66    |
> > > | CodeLlama-34b | 497    | 163  | 102 | .80  | .79 | .83   | .79    |
> > > | Llama-70B     | 280    | 380  | 18  | .70  | .59 | .94   | .42    |
> > > | GPT-4         | 292    | 368  | 13  | .71  | .61 | .96   | .44    |
> > >
> > > | Slicing Method   | Detect | Miss |
> > > | ---------------- | ------ | ---- |
> > > | Def-use + Window | 497    | 163  |
> > > | Window Only      | 531    | 129  |
> > > | Def-use Only     | 491    | 169  |
> > > | Thin Slicing     | 452    | 208  |
> > > | Full Slicing     | 438    | 222  |

---

> > > > ### Author Response · Authors · 2025-06-09
> > > > **Follow-up**
> > > >
> > > > Thank you again for your thoughtful and detailed review. In our rebuttal, we’ve addressed your concerns regarding dataset construction (including why we focused on edges missed by ACG), the use of negative samples, the completeness of our TodoMVC traces, and the reported numbers in Table 2/4 and the abstract.
> > > >
> > > > If you have any remaining questions or would like to follow up on any point, we’d be happy to clarify further!

---

### Official Review · Reviewer_JHym · 2025-05-23

**Rating:** 9
**Confidence:** 4
**Ethics Flag:** 1

**Summary:**

CALLME addresses a longstanding weakness of JavaScript static analysis: resolving calls made through dynamic-property access (eg `obj[prop]()` ). It does so in two phases:

1. JSelect builds a slice of context that captures only what an analyst would read.

2. The slice and a candidate callee are fed to an LLM (best results with Code-Llama-34 B) via an “abstract-interpretation” prompt that asks, “Does this call invoke function F?”

On 660 challenging edges from the TodoMVC benchmark, CALLME improves recall from 11 % → 75 % and F-score from 0.19 → 0.79 over the best previous static tool (Jelly). A case study of 25 real vulnerabilities shows 24 detcted with only 3 false positives.

**Questions To Authors:**

1. How will candidate callees be generated at scale without exploding the number of LLM queries?

2. What, if anything, is special about the remaining 25 % of missed edges?

Presentation: please increase the resolution of Figure 3! Its current blurriness undermines an otherwise excellent paper.

**Reasons To Accept:**

Solid work: strong results. The hybrid slice-plus-LLM design shows an order-of-magnitude accuracy boost that immediately improves downstream security and refactoring tools.

**Reasons To Reject:**

None. I see no substantive flaws.

---

> ### Author Response · Authors · 2025-06-03
> **Response to question 1 on generating candidate callees**
>
> Thank you for your kind comments, we are glad you enjoyed the paper! We have responded to your questions below :)
>
> > How will candidate callees be generated at scale without exploding the number of LLM queries?
>
> CALLME is designed for **selective use**, targeting only statically identified dynamic property accesses that traditional analysis cannot resolve. This is tractable because many downstream analyses operate on-demand, requiring only specific call edges. For example, in **taint analysis**, the goal is to determine whether data flows from a source to a sink; in **dead code elimination**, we need to know whether a function is reachable via dynamic dispatch; and in **static security auditing**, an analyst may ask whether a sensitive call site (e.g., `evalHook`) is ever reached dynamically. In all of these cases, resolving a small number of dynamic calls is far more important than recovering the entire call graph. Therefore, these downstream tasks typically do not incur many LLM queries. For example, in our case study for taint analysis, it incurs 3.79 calls to the LLM on average.

---

> > ### Author Response · Authors · 2025-06-03
> > **Response to question 2 on missed edges**
> >
> > > What, if anything, is special about the remaining 25% of missed edges?
> >
> > We performed an error analysis of the ~25% of missed edges (162 total) and categorized them. **Note that one missed edge can have multiple possible errors, which is why the “# missed” sums to more than 162.**
> >
> > | Causes                                         | # missed |
> > | ---------------------------------------------- | -------- |
> > | Missing context due to limited code extraction | 63       |
> > | Function arguments not resolved                | 101      |
> > | Other                                          | 11       |
> > **Missing context due to limited code extraction (63 cases)**: In these cases, our static preprocessing failed to extract enough relevant code for the model to make an informed prediction. This typically happens when the call site depends on values defined through a long chain of reassignments or spread across a large object that is built up dynamically. JSelect only follows one level of variable usage, which sometimes causes important information to be excluded from the model input. In fact, we implemented tracing multiple levels of variable usage in JSelect, but it came at the cost of significantly increased context length for LLMs (approximately 3x). Studying how to balance precision (with richer context) and scalability (LLM inference support) can be interesting in future works.
> >
> > **Function arguments not resolved (101 cases)**: Most failures involve call sites where the dynamic field depends on a function argument. Resolving this would require analyzing all possible call sites of that function, which is nontrivial and leads to a chicken-and-egg problem, i.e., `f` in `obj[f]` is a parameter to the parent function. Resolving f requires knowing the call graph, but constructing the call graph requires resolving all the calls. We have acknowledged this case in the paper in Section 3.1, where we discussed JSelect’s limitation of not being interprocedural.
> >
> > **Other (11 cases)**: Includes cases such as unintuitive naming of fields or just a straightforward LLM misprediction.

---

> > ### Comment · Reviewer_JHym · 2025-06-03
> > **Thanks for your response**
> >
> > > Missing context due to limited code extraction (63 cases): In these cases, our static preprocessing failed to extract enough relevant code for the model to make an informed prediction.
> >
> > This is indeed interesting future work!
> >
> > Also worth noting that only 11 cases potentially involve a limitation directly attributable to the LM used in this work; that's exciting and a reason to explore this problem further with even more ambitious questions.

---

> > > ### Author Response · Authors · 2025-06-09
> > > **Follow-up**
> > >
> > > Thank you so much! We really appreciate your comments. It’s exciting for us as well, and we’re looking forward to exploring those future directions. If you have any further questions or suggestions, we’d be very happy to follow up!

---

> ### Comment · Reviewer_JHym · 2025-06-09
> **Nit**
>
> (repeating from my review) Presentation: please increase the resolution of Figure 3! Its current blurriness undermines an otherwise excellent paper.

---

> > ### Author Response · Authors · 2025-06-10
> > **Response**
> >
> > Thank you, we will definitely increase the resolution in our revision!

---

### Author Response · Authors · 2025-06-10
**Summary Response**

Dear Area Chairs and Program Chairs,

Thank you for handling the discussion and for managing the review process. We appreciate the opportunity to present our work to the COLM community. In the following, we summarize the key contribution of our paper and how we addressed the reviewers’ questions and concerns during the discussion phase.

__CALLME’s core contribution__ is to resolve dynamic property accesses in JavaScript using LLMs, filling a key blind spot in static call graph construction. Our approach augments existing static analyses by selectively querying an LLM when traditional techniques fail, and we evaluate this on both a call resolution benchmark and a downstream taint analysis case study.

In response to the reviewers’ questions and feedback, we have added the following clarifications and new contextual explanations:

1. __Error analysis (Reviewer JHym)__: We added a detailed error analysis of the 162 missed cases in our call resolution benchmark. We found that most failures were due to __missing context from limited static extraction (63 cases)__ and __unresolved function arguments (101 cases)__, which highlights actionable opportunities for improving preprocessing and LM performance.
2. __Bug detection evaluation (Reviewers NECr, 7TUv)__: We clarified the construction of the bug detection dataset and __added a new experiment__ that Jelly, the current state-of-the-art static analyzer for JavaScript, __fails to detect any of the 24 vulnerabilities__. This underscores the practical value of CALLME in handling cases that existing static analyses miss entirely.
3. __Negative edge construction rationale (Reviewers NECr, 7TUv)__: We clarified that CALLME is framed as a __binary decision tool__—given a dynamic call site and a candidate callee, should an edge exist? The use of randomly selected functions as negatives reflects realistic downstream settings (e.g., taint tracking, auditing), where the task is to evaluate specific call edges. These are __not artificially weak negatives__, but plausible alternatives that mirror real-world analyst uncertainty.
4. __Clarification on candidate selection methodology (Reviewers vcVB, 7TUv)__: We clarified that the process for selecting candidate callee nodes is __task-dependent__ and handled by upstream components. CALLME is designed to operate on candidates provided by the downstream analysis.

Thank you again for managing the discussion and for your oversight throughout the process!

---

### Decision · Program_Chairs · 2025-07-08

**Decision:**

Accept

**Comment:**

This paper introduces CALLME, a hybrid approach that combines large language models (LLMs) with static analysis to resolve dynamic property accesses in JavaScript, a longstanding challenge for static call graph construction. The authors demonstrate that CALLME significantly outperforms state-of-the-art static analyzers by using a lightweight statement selection technique (JSelect) to build relevant context before querying an LLM. The evaluation shows impressive gains in both a call resolution benchmark and a downstream security vulnerability detection case study. The work represents a meaningful contribution to the emerging field of neuro-symbolic program analysis, demonstrating how LLMs can extend the capabilities of traditional static tools in areas where purely symbolic reasoning breaks down due to dynamic language features.

Pros:
- Addresses a well-defined, challenging problem in JavaScript static analysis that traditional symbolic methods struggle with
- Shows significant performance improvements over state-of-the-art static analyzers (75% recall vs 11% for Jelly)
- Demonstrates practical utility in downstream tasks like vulnerability detection

Cons:
- Limited evaluation scope focused specifically on dynamic property accesses
- Evaluation methodology presents CALLME as a binary classifier rather than demonstrating integration into a full call graph construction pipeline
- Small-scale case study for vulnerability detection (24 examples)